# miR-146a, miR-221, and miR-155 are Involved in Inflammatory Immune Response in Severe COVID-19 Patients

**DOI:** 10.3390/diagnostics13010133

**Published:** 2022-12-30

**Authors:** Noemí Gaytán-Pacheco, Alejandro Ibáñez-Salazar, Ana Sofía Herrera-Van Oostdam, Juan José Oropeza-Valdez, Martín Magaña-Aquino, Jesús Adrián López, Joel Monárrez-Espino, Yamilé López-Hernández

**Affiliations:** 1Clinical Analysis Laboratory UAZ-Siglo-XXI, Academic Unit of Chemical Sciences, Autonomous University of Zacatecas, Zacatecas 98000, Mexico; 2Faculty of Medicine, Autonomous University of San Luis Potosí, San Luis Potosí 78210, Mexico; 3Metabolomics and Proteomics Laboratory, Academic Unit of Biological Sciences, Autonomous University of Zacatecas, Zacatecas 98600, Mexico; 4Central Hospital Dr. Ignacio Morones Prieto, San Luis Potosí 78210, Mexico; 5MicroRNAs and Cancer Laboratory, Academic Unit of Biological Sciences, Autonomous University of Zacatecas, Zacatecas 98000, Mexico; 6Department of Health Research, Christus Muguerza del Parque Hospital Chihuahua, University of Monterrey, San Pedro Garza García 66238, Mexico; 7CONACyT-Metabolomics and Proteomics Laboratory, Autonomous University of Zacatecas, Zacatecas 98000, Mexico

**Keywords:** miRNAs, SARS-CoV-2, immune response

## Abstract

COVID-19 infection triggered a global public health crisis during the 2020–2022 period, and it is still evolving. This highly transmissible respiratory disease can cause mild symptoms up to severe pneumonia with potentially fatal respiratory failure. In this cross-sectional study, 41 PCR-positive patients for SARS-CoV-2 and 42 healthy controls were recruited during the first wave of the pandemic in Mexico. The plasmatic expression of five circulating miRNAs involved in inflammatory and pathological host immune responses was assessed using RT-qPCR (Reverse Transcription quantitative Polymerase Chain Reaction). Compared with controls, a significant upregulation of miR-146a, miR-155, and miR-221 was observed; miR-146a had a positive correlation with absolute neutrophil count and levels of brain natriuretic propeptide (proBNP), and miR-221 had a positive correlation with ferritin and a negative correlation with total cholesterol. We found here that CDKN1B gen is a shared target of miR-146a, miR-221-3p, and miR-155-5p, paving the way for therapeutic interventions in severe COVID-19 patients. The ROC curve built with adjusted variables (miR-146a, miR-221-3p, miR-155-5p, age, and male sex) to differentiate individuals with severe COVID-19 showed an AUC of 0.95. The dysregulation of circulating miRNAs provides new insights into the underlying immunological mechanisms, and their possible use as biomarkers to discriminate against patients with severe COVID-19. Functional analysis showed that most enriched pathways were significantly associated with processes related to cell proliferation and immune responses (innate and adaptive). Twelve of the predicted gene targets have been validated in plasma/serum, reflecting their potential use as predictive prognosis biomarkers.

## 1. Introduction

SARS-CoV-2 virus is transmitted through aerosols and droplets. So far, the COVID-19 pandemic has resulted in more than 530 million cases and 6.3 million deaths [1].

Upon transmission, the virus particles bind the lung epithelial cells through interactions between the spike (S) protein and the host cellular entry receptor angiotensin-converting enzyme 2 (ACE2) [2], causing a negative regulation of ACE-2 with the consequent loss of its catalytic capacity to degrade angiotensin II. During the infectivity phase, the virus can also produce its own miRNAs, which could further regulate host miRNAs and their targets [3]. These host miRNAs may influence different phases of the viral life cycle, including translation, by attaching itself to the viral RNA or mRNA.

Viruses can induce the up-/downregulation of certain host miRNAs to evade the host’s immune system by suppressing antiviral factors, such as interferon (IFN) [4,5,6]. On the other hand, several miRNAs could have an antiviral effect, enabling the defense mechanisms to fight the infection [5,7].

Previous data on the mechanisms regulated by miRNAs suggest a possible role in COVID-19; miRNAs for which significant alterations have been reported are implicated in the regulation of the immune and/or inflammatory pathways at different levels: cytokine and chemokine synthesis; T-cell development, differentiation, and activation; or B-cell development, differentiation, and activation, among others [8]. Well-described associations have been reported in ventilated patients between miR-155 and inflammation; miR-208a/miR-499 and myocardial/cardiomyocyte damage; and miR-21/miR-126 in cardiac fibroblast and endothelial-cell dysfunction [9].

In this context, the use of molecular principles of gene regulation mechanisms could represent an innovative way to identify potential biomarkers of infection for developing antiviral therapeutic agents for certain diseases that do not have yet an effective treatment, such as COVID-19. Despite there are validated and approved methods available for COVID-19 diagnosis, prognosis is still complicated by only assuming the positive results of rapid antigen tests, serological tests, or RT-qPCR. For this reason, we are continuously looking for reproducible methods that are able to predict poor prognosis even at early points after infection. Most COVID-19 studies have focused on proteomic, metabolomic, and cellular biomarkers [8]. In the last decade, noncoding RNAs (ncRNAs) and miRNAs, have emerged as novel tools to aid in medical decision-making and can be easily measured through standard techniques already employed in clinical laboratories, such as RT-qPCR. miRNAs are sensitive, robust, and cost-effective biomarkers that offer additional information to already established clinical variables and clinical indicators [10]. They are stable in various body fluids and offer advantages as biomarkers because they are highly conserved between species, and their expression patterns are tissue and life-stage specific. The advantage of miRNAs is their consistent detectability in patients with lower COVID-19 severity and a lesser dependence on sampling time. The integration of miRNAs into biomarker signatures may improve the performance of established biomarkers, as demonstrated for binary and triplet combinations with D-dimer, troponin T, SARS-CoV-2, RNAemia, age, and BMI [11]. Target genes for these miRNAs play an important role in the immune dysregulation observed in COVID-19 patients, and they may also be evaluated as potential biomarkers in the context of long-COVID, because cumulative evidence suggest immune alterations in patients with persistent symptoms or post-COVID-19 symptoms.

For the aims of the present work, we chose five miRNAs with previously validated involvement in inflammatory processes related to immune system activation [12,13,14,15] to examine their expression in plasma from patients with severe COVID-19 and to explore their role as potential biomarkers of disease severity. It is well acknowledged that hyperinflammation and massive cytokine dysregulation are mechanisms leading to poor outcomes in severe COVID-19 patients [16]. Due to the heterogeneity of the factors that could affect the course of the disease, we propose here an age- and gender-adjusted model to differentiate controls from severe COVID-19 patients.

## 2. Materials and Methods

### 2.1. Study Population and Sampling

This is a cross-sectional study with 41 patients suffering from severe COVID-19 and 42 negative controls attending hospitals for COVID-19 diagnosis or treatment. SARS-CoV-2 diagnosis was conducted using reverse transcription polymerase chain reaction (RT-PCR) from a nasopharyngeal specimen using standard methods [17,18].

This study was revised and approved by the Ethics and Research Committees of the Christus Muguerza del Parque Hospital (folio: CEI-HCMP-15042020-3) and Health Secretary Services of San Luis Potosí (folio: CEI-003-20161034). The study was conducted in accordance with the Declaration of Helsinki. Informed consent was obtained from all participants prior to the collection of the blood samples. Figure 1 shows the workflow for the experimental design and data analysis.

### 2.2. miRNAs Selection

One of the preferred methods to study miRNAs is microarrays, which requires a large amount of RNA sample (usually more than 1 µg). Since COVID-19 has been widely acknowledged as a hyperinflammatory diseases with a strong dysregulation in the immune system, it is remarkable to find the factors that control this inflammatory unbalance. For this reason, more than to perform a massive screening of all miRNAs, we selected some miRNAs that have been previously related to inflammation, immune response, vascular complications, metabolic signaling, and organ damage. Based on recent bioinformatic prediction studies [11,19], we selected five miRNAs associated with some of the processes previously mentioned and ranked within the most important miRNAs. Our goal was to validate previous bioinformatic predictions for these miRNAs, this time for a Mexican population, which remains poorly explored until now. Appendix A shows the miRNAs selected for validation and the justification for their inclusion in our study based on previous findings.

### 2.3. Circulating miRNA Isolation and Relative Expression Determination by RT-qPCR 

Total RNA from each sample was isolated from 25 μL plasma using TRI-Reagent (Sigma-Aldrich, Germany), according to manufacturer’s instructions. The isolated RNA was resuspended in DEPC-treated and RNAse inhibitors. Then, the RNA quality was measured at 260/280 nm using a spectrophotometer (Q3000 Quawell Technology, Inc., San José, CA; USA). A value greater than 1.8 was considered acceptable. The cDNAs for the mature miRNAs (U6 snRNA, miR-16-5p, miR-221-3p, miR-34-5p, miR-146a, and miR-155) were synthesized from 300 ng of total RNA by one-step RT-qPCR using the GoldBio’s Probe-Based One Step RT-qPCR Kit (Gold Biotechnology^®^ St Louis, MO; USA); TaqMan probes were used for each of the miRNAs to avoid unspecific amplification. The miRNA amplification by qRT-PCR was carried out using TaqMan MicroRNA Assay specific primers (Applied Biosystems, Foster City, CA; USA) in a thermocycler (qTOWER^3^; AnalytikJena, Gottingen, Germany) with the following amplification conditions: first strand cDNA synthesis at 42 °C for 30 min, initial denaturation/RT inactivation at 95 °C for 3 min, followed by 40 cycles of denaturation at 95 °C for 5 s, annealing/extension at 60 °C for 30 s. 

All RT-qPCR reactions were performed in duplicate. Cq values were averaged and the ΔΔCq method [20] was used to obtain the relative expression, where ΔΔCq was calculated by subtracting the ΔCq value from the mean of the control group with the ΔCq value of the COVID-19 patients. NormFinder [21] was used to estimate the best candidate reference gene between U6 snRNA and RNU48; miRNA levels were normalized with the use of the average of reference Cq value as the housekeeping gene, according to NormFinder parameters.

### 2.4. Bioinformatic Analysis

Bioinformatic analysis of targets was performed through the page miRNet 2.0 and its web service https://www.mirnet.ca/miRNet/home.xhtml (accessed on 14 July 2022). This analysis identified targets that were shared by 2 or more miRNAs. We assessed hsa-miR-155-5p, hsa-miR-221-3p, and hsa-miR-146a-3p, which were the significant miRNAs. The selection of most relevant genes was made, setting degree centrality and betweenness to equal or >1. Using the same web service, a miRNA-gene network with selected targets was constructed. Subsequently, all genes related to immune system functions were selected. Functional association analysis using the TAM 2.0 http://www.lirmed.com/tam2/ (accessed on 17 July 2022) [22] was also conducted to identify the functional terms for differentially expressed miRNAs. To control for multiple comparisons, a false-discovery rate (FDR) < 0.05 was used.

### 2.5. Statistical Analysis

Frequencies and proportions were used to describe sociodemographic characteristics, main comorbidities, symptomatology, and epidemiological data for the study participants. 

Statistical analyses were carried out using GraphPad (version 5.0) software (GraphPad, La Jolla, CA, USA). The mean ± SD or median ± IQR were used to represent continuous data with parametric or nonparametric distribution, respectively. For clinical data, ANOVA with Tukey’s post hoc tests for continuous variables were used to identify differences across categories, and Fisher’s exact tests were used for nominal data. Nonparametric Kruskal–Wallis with Dunn’s post hoc tests were employed to identify differences in miRNA data. Statistical significance was set at *p* < 0.05.

Spearman’s correlation coefficients and plots between miRNAs and laboratory values were computed using R studio (4.1.2).

Crude and adjusted logistic regression models were built to predict COVID-19 severity. Odds ratios (OR) with 95% confidence intervals (CI) were computed. The full model included all variables, with a *p* ≤ 0.10 in crude analyses, but only independent variables with a *p* < 0.05 in at least one category in the comparisons remained in the final model. A receiving operating characteristics (ROC) curve was produced from the final logistic model, and the area under the curve (AUC) was reported. The Nagelkerke pseudo-R^2^ statistic, ranging from 0 to 1, was used to provide an indication of the amount of variation in the dependent variable explained by the model.

## 3. Results

### 3.1. Sociodemographic Characteristics

The sociodemographic characteristics of the patients recruited in the present study are summarized in Table 1. The mean age of the patients with severe COVID-19 was higher than that of healthy controls (52.4 vs. 39.1 years, *p* < 0.01). Male patients with severe COVID-19 accounted for 82.9%, compared with 45.2% in the control group (*p* < 0.01). In terms of comorbidities, the proportion of patients with diabetes (26.8% vs. 2.3%; *p* < 0.01) and hypertension (41.4% vs. 9.5%; *p* < 0.01) was higher among those with severe COVID-19, but no statistical differences were seen for the other conditions assessed. All general and respiratory symptoms measured were clearly more frequent (*p* < 0.01) in patients with severe COVID-19 than among controls.

### 3.2. Differential Expression of miRNAs in the Plasma of Patients with COVID-19

Figure 2 shows the relative expression (normalized against U6 snRNA) in controls and COVID-19 patients for each studied miRNA. Significant upregulation of miR-16, miR-155, and miR-221 was observed for COVID-19 patients (Figure 2). miR-146a was found to be marginally significant in the crude analysis. NormFinder analysis identified U6 snRNA as the single most stable gene, with a stability of 0.589 and a standard error of 0.33.

### 3.3. Logistic Regression Model Based on miRNAs for Classification of COVID-19 Patients

Crude and adjusted regression models (OR; 95% CI) to differentiate patients with COVID-19 from healthy controls are presented in Table 2. The results showed a statistically significant upregulated expression of miR-155, miR-16, and miR-221 in patients with severe COVID-19 compared with healthy controls in crude analyses. However, when adjusting by age and sex, only miR-155, miR-146a, and miR-221 remained significant for identifying COVID-19 patients.

### 3.4. Model Performance

Figure 3A represents the ROC curve for all the individual miRNAs, without adjustment. However, when adjusted by sex and age, the combination of miR-155, miR-146a, and miR-221 showed an adequate performance (AUC: 0.95, 95%CI 0.89–0.98) (Figure 3B) for discrimination of severe patients.

### 3.5. Correlation of Significant miRNAs with Clinical Variables in Severe COVID-19 Patients

To assess if differences in miRNA expression were related to other variables, we performed a correlation analysis. There was a significant positive association between miR-146a and absolute neutrophil count (r = 0.57, *p* = 0.007), and proBNP (r = 0.40, *p* = 0.0001), also between miR-221 and ferritin (r = 0.35, *p* = 0.03). On the other side, we observed a significant negative correlation (r =−0.61, *p* = 0.01) between miR-221 and total cholesterol (Figure 4).

### 3.6. Evaluation of the Interaction Networks of Common Target Genes across the Studied miRNAs

Only validated target genes with a correlation ≥1 were entered in the analysis to identify genes regulated by miR-155-5p, miR-221-3p, and miR-146a-3p, which were the miRNAs that contributed to the adjusted logistic regression model. A total of 498 targets were related to the analyzed miRNAs (Figure 5A). The Venn diagram (Figure 5B) shows the number of genes shared by the miRNAs under study. These genes are listed in Table 3.

From the 498 validated targets, 12 genes have been validated for miRNAs regulation in plasma/serum. Table 3 shows the validation method, as well as the function associated with each target gene. These genes are the most important target genes to be considered as potential biomarkers since previous studies have validated them in the same matrix in which miRNAs were measured in our study.

Finally, in the functional association analysis (Figure 6), miRNAs statistically different in COVID-19 patients showed enriched functional terms, including aging, apoptosis, T-cell differentiation, hematopoiesis, and immune response, among others.

## 4. Discussion

This study looked at the differential expression of five circulating miRNAs in plasma from healthy controls and patients with severe COVID-19 to explore their potential role in the inflammatory and host immune response against infection, and to assess their value as biomarkers of disease severity. Our group has been focused in the role of miRNAs as regulators for certain diseases [33,34,35]. Evidence suggests that these molecules are important predictors of disease complications and an important source of diagnosis biomarkers or therapeutic targets. The Mexican population is complex to study, since multiple factors, including ethnicity, comorbidities, genetic factors, and lifestyle, could modify disease outcomes when comparing with other populations. We have previously seen these miRNAs modulated in other inflammatory diseases [36], and we aimed to find a link between COVID-19 and all the inflammatory mediators that influence the hyperinflammatory status observed in severe cases. Therefore, with the present study, we validated not only our previous findings about the involvement of the studied miRNAs in inflammation and immune system, but also, we validated other author contributions, demonstrating that, independently of population differences, these miRNAs regulate important processes related to COVID-19 complications. Analysis of the identified miRNAs showed regulatory functions associated with inflammation, immune response, and vascular and metabolic diseases, indicating that the infection caused by the SARS-CoV-2 virus has multiple effects that alter the homeostasis of different organs and tissues.

Previous studies have shown that viral infections affect host homeostasis by regulating miRNA expression [11,37,38,39,40,41]. The altered expression of miRNAs causes other genes to regulate the host immune response to viral infection. Each miRNA can then target multiple genes, making them important regulators of numerous cellular functions. Eyileten et al. (2022) reported a bioinformatic miRNA prediction and subsequent validation in Poland patients. The authors looked for miRNAs regulating the highest number of top network-medicine-based integrative approach (NERI) nodes and top NERI targets associated with coagulation. In their study, top miRNAs were identified based on their regulation of the highest number of the top differentially expressed genes associated with coagulation and involved in the coagulation process. The miRNAs selected in our study are ranked in the top 30 miRNAs having a role in ACE2-related thrombosis in coronavirus infection [19]. 

The results presented here showed a statistically significant dysregulated expression of miR-155, miR-16, and miR-221 in patients with severe COVID-19 compared with healthy controls in crude analyses. However, for the establishment of an adequate predictive model, adjustment for cofounders must be accomplished. Due to the presence of different factors influencing COVID-19 outcomes, such as sex, age, and comorbidities, we propose here an adjusted model. When adjusting by age and sex, only miR-155, miR-146a, and miR-221 remained significant for identifying severe COVID-19 patients. ROC built with the adjusted model showed an adequate performance (AUC: 0.95, 95% CI 0.89–0.98). However, interpretation of these results must be done with caution. A low sensitivity was achieved in our study. Sensitivity refers to the proportion of subjects who have the target condition (reference standard positive) and give positive test results. This parameter is highly dependent on the simple size. For the sample size calculation, the prevalence of the target population must be considered to obtain a representative sample. With a low or high prevalence, the study may be overpowered in one subpopulation. Since COVID-19 was an emergent disease, these estimations were difficult to calculate at the beginning of the pandemic. Sammut-Powell et al. [42] reported a simulation study to evaluate the effect of sample size calculations in the sensitivity and specificity of COVID-19 diagnostic tests in practice. Under the current emergency guidelines from the Medicines and Healthcare Products Regulatory Agency, companies are required to evaluate diagnostic tests in 30 positive and 30 negative cases. The authors demonstrated that, in practice, in a test performed with 80% sensitivity and 93% specificity in 30 positive and negative samples, respectively, their real-world sensitivity and specificity could be as low as 57.7% and 83.2%, respectively. 

Our results are consistent with previous studies in which expression of miR-155 has been increased in patients with COVID-19 [43,44,45]. A study by Garg et al., found overexpressed values of miR-155 in critically ill and mechanically-ventilated COVID-19 patients, suggesting that this miRNA could be useful for evaluating the severity of the disease [9]. This overexpression seems to contribute to the overall exacerbated proinflammatory state widely described in patients with COVID-19. miR-155 has its origin in leukocyte cells, and there is increasing evidence that macrophage activation contributes to the initiation of inflammatory responses resulting in tissue damage [23]; miR-155 responds to many inflammatory stimuli, such as TNF-α, IL-1β, pathogen-associated molecular patterns (PAMPs), and damage-associated molecular patterns (DAMPs) that act by potentiating the inflammatory response [15]. Additionally, miR-155 has an important role in innate immunity and differentiation and activation of T and NK cells [15,23,45,46]. miR-155-5p also has an important role in inflammatory and immunological processes, endothelial dysfunction, and cardiometabolic diseases. Moreover, it has been shown that miR-155 upregulates IL-1, IL-6, TNF-α, and IL-12 signaling pathways, as well as the NF-kB and JAK/STAT3 pathways. Therefore, over-regulation of miR-155 in SARS-CoV-2 infection could be a good predictor of inflammatory status and immune disorders [9,43,44,47,48,49]. Studies have found that this miRNA is also differentially expressed in other conditions, including acute lung damage, viral infections (influenza), pulmonary fibrosis, and asthma [44,50,51,52].

On the other hand, miR-221-3p is able to target molecules belonging to pathways with key roles in inflammatory responses, including toll-like receptors (TLRs), transcription factors (NF-kB), and cytokines/chemokines (TNF-α, IL-6, and IL- 8) [53,54,55]. miR-221 has been differentially expressed downwards in patients with COVID-19 compared to those with community-acquired pneumonia [39]. Molinero et al. found an overexpression in the ratio of miR-221-3p in bronchial pulmonary aspirate samples in patients seriously ill with COVID-19, compared with healthy controls (in which decreased levels of miR-221-3p in non survivors were reported), suggesting that the signature of miRNAs may change depending on the degree of severity, the type of tissue studied, and the outcome of the disease [44]. The proposed miR-221-3p mechanisms may be severe endothelial injury and coagulopathy (features observed in lung samples of fatal COVID-19 cases) [56] to the targeting core receptor protein ADAM17 (a disintegrin and metalloproteinase 17) involved in ACE2-dependent shedding (associated with lung pathogenesis) [57], and to the suppression of the innate immune response and promotion of the viral infection via the TBK1 (TANK binding kinase 1) gene [58,59].

Previous experimental studies have described downregulation of miR-146a during SARS-CoV-2 infection [50,53,60]. miR-146a is known to be an anti-inflammatory miRNA. It regulates inflammation by targeting the factor 6-associated TNF receptor (TRAF6), thereby reducing the expression of NF-kB (nuclear factor kappa B) [12,61,62]. miR-146a depletion leads to IL-1, IL-6, and TNFα overproduction [63]. It is also related to the toll-like receptor signaling pathway and is a regulator of IL-1B and TGFB1 transcription factors [64,65].

miR-146a expression also correlated with biochemical parameters such as proBNP and absolute neutrophil count. Natriuretic peptides are sensitive indicators of cardiac and hemodynamic stress, which may be due to left ventricular systolic/diastolic dysfunction, ischemic or inflammatory dysfunction, and right cardiac overload secondary to pulmonary consequences of the disease (i.e., pulmonary embolism, pulmonary hypertension, hypoxic vasoconstriction, or acute respiratory distress syndrome) [66]. In fact, the potential usefulness of this cardiac parameter as a prognostic biomarker associated with the severity of COVID-19 has been previously suggested [67,68]. Neutrophils are at the intersection of innate immune responses, including pathogen destruction, thrombosis, and activation of the adaptive immune system [69,70].

Lastly, miR-146a directly regulates thromboinflammatory processes by inhibiting several proinflammatory elements of the NF-κB pathway [13,71]. This miRNA is predominantly expressed in cells that promote thrombosis (i.e., macrophages, platelets, neutrophils, and endothelial cells) [72]. In addition, it has been shown that decreased levels of miR-146a in patients with pneumonia are associated with an increased risk of adverse cardiovascular events by exacerbating the inflammatory and prothrombotic responses associated with severe COVID-19 [70].

We also account for a positive correlation between ferritin and miR-221. It has been reported that ferritin is a nonspecific marker of inflammation and a key mediator of immune dysregulation through direct immunosuppressive and proinflammatory effects that contribute to the cytokine storm [73]. Clinically, cytokine release storms are a common phenomenon in patients with SARS-CoV-2. This process results in multiple deleterious effects on both the innate and acquired immunity, potentially related to the activation and differentiation of the T-cellular process wherein miRNAs have essential functions in various immune-related diseases and could therefore modulate the response during viral infections.

The negative correlation between total cholesterol and miR-221 (r = −0.61) is also relevant, as lipid metabolism plays an essential role in the COVID-19 disease. Cholesterol has been shown in several studies to interact with the S protein of SARS-CoV-2 [74]. Decreased serum total cholesterol levels have been associated with poor prognosis in patients with COVID-19. Ressaire et al., reported that low total cholesterol levels could result from SARS-CoV-2-induced vasculopathy; the authors also observed a positive correlation between total blood cholesterol levels and COVID-19 severity, which was evaluated using the Kirby index [75].

Functional analysis showed that most enriched pathways were significantly associated with processes related to cell proliferation and immune responses (innate and adaptive). CDKN1B was found to be the unique shared target between the significant miRNAs. This gene has an important role controlling the cellular cycle and apoptosis. A recent study found that p27Kip1, encoded by CDKN1B, was positively regulated by the innate immune signaling activated by the Influenza A virus. The authors suggested that increased expression of p27Kip1 could limit the viral replication, constituting a potential therapeutic approach [76].

In conclusion, SARS-CoV-2 infection appears to induce an important response in the host’s miRNA profile, suggesting that the severity of the symptoms is associated with epigenetic factors, which in turn regulate a large number of functions in different tissues that can modulate the host response. The differential expression of miRNAs involved in inflammatory processes, including miR-155, miR-221, and miR-146a found in patients with severe COVID-19 in this study, point to their potential role as regulators of cellular processes in SARS-CoV-2 infections. Therefore, the expression levels of these dysregulated miRNAs could be of diagnostic and prognostic value as biomarkers to predict the severity of the disease and to develop therapeutic strategies against the virus through their regulation. To our knowledge, this is the first study reflecting miRNA dysregulation in severe Mexican COVID-19 patients and their association with inflammation, immune system, and vascular complications leading to organ failure. Since several studies about miRNA dysregulation are based on bioinformatic prediction, the validation of these findings, such as we are presenting here, will contribute to the inclusion of these molecules in multiomics prognostic panels. However, we need to acknowledge study limitations. Sample size was the major limitation, which makes it conflictive to establish highly accurate models of prediction. Moreover, the sample amount (volume) limited the validation of targets by traditional methods such as Western blot.

For future studies, the sample size needs to be increased, including patients with mild COVID-19. miRNAs could also be evaluated in survivors and non survivors, as well as in other types of samples that could be less invasive, such as urine, sputum, saliva, etc.

## Figures and Tables

**Figure 1 diagnostics-13-00133-f001:**
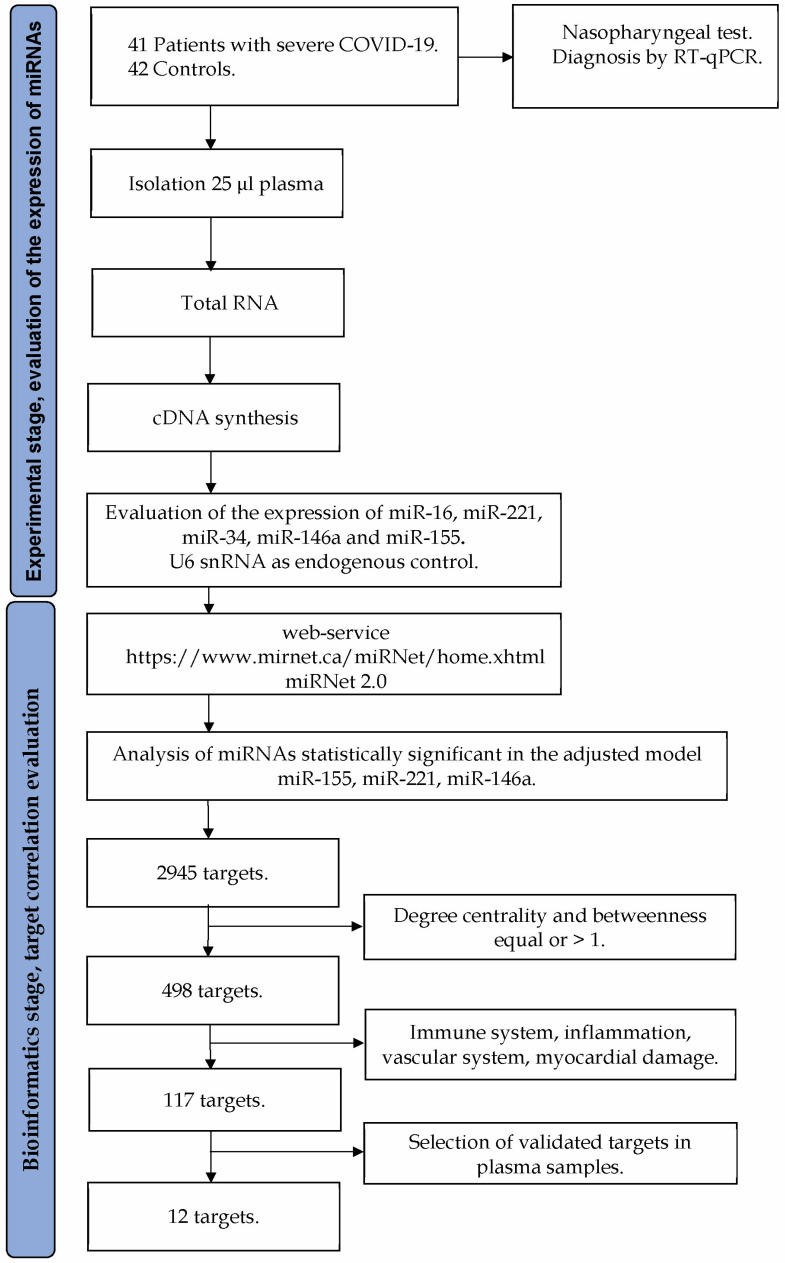
Flowchart diagram of the experimental design and data analysis.

**Figure 2 diagnostics-13-00133-f002:**
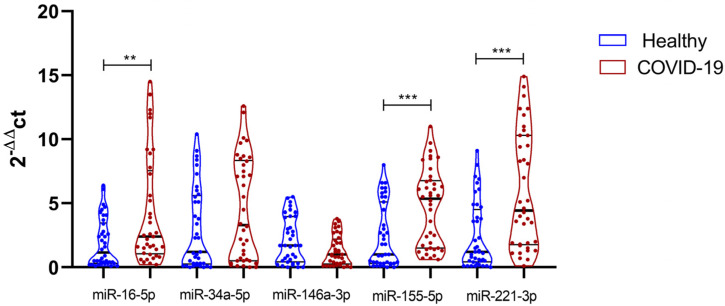
Violin plot shows the relative expression of miRNAs in COVID-19 patient vs. healthy controls. The violins represent medians and interquartile ranges. Data were analyzed using the Kruskal–Wallis test and Dunn’s post hoc test when significant values were obtained (** and *** *p* ˂ 0.05).

**Figure 3 diagnostics-13-00133-f003:**
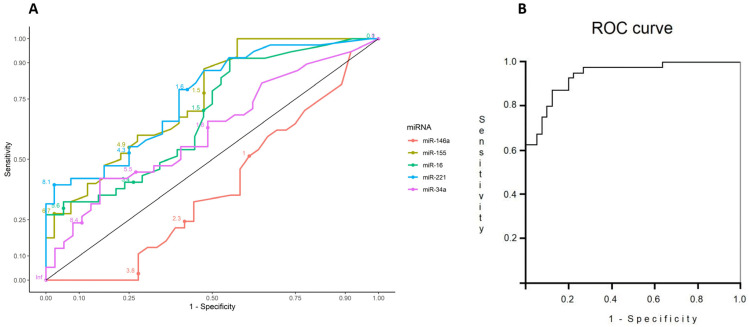
(**A**) ROC curve with individual miRNAs. (**B**) ROC curve built with the variables included in the logistic regression model. (AUC: 0.95, 95%CI 0.89–0.98). Nagelkerke pseudo-R^2^ statistic, ranging from 0 to 1.

**Figure 4 diagnostics-13-00133-f004:**
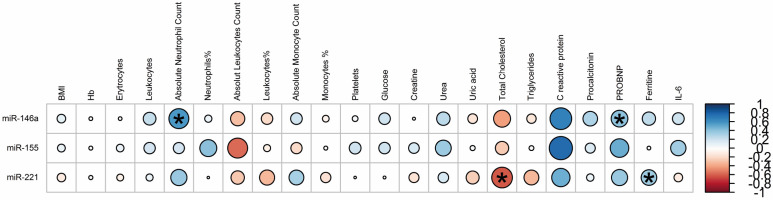
Correlations between significant miRNAs and clinical characteristics. Blue circles represent positive correlations, while red circles represent negative correlations. Significance of correlation is, accordingly, the circle diameter. Spearman’s correlation (* *p* < 0.05).

**Figure 5 diagnostics-13-00133-f005:**
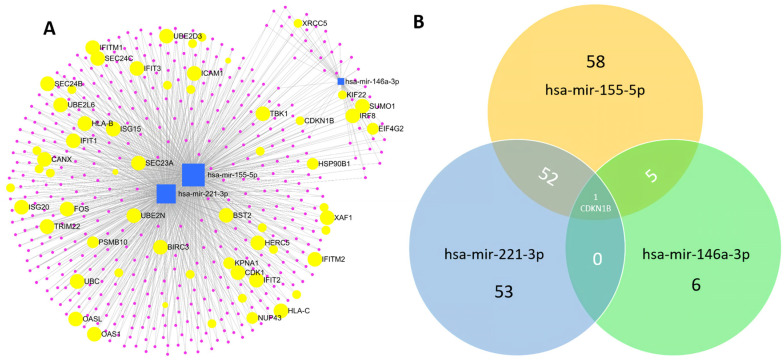
miRNA-target gene network related to immune system. (**A**) Results of pathway and network analyses for the 3 miRNAs, selected according to the logistic regression model. Blue squares indicate miRNAs, pink dots indicate corresponding targets, and yellow circles represent genes involved in immune system functions that can be targeted by two or more of the selected miRNAs. (**B**) Venn diagram shows the miRNAs 155-5p, 146a-3p, and 221-3p and the number of their immune system targets. CDKN1B gene is shared between miRNAs.

**Figure 6 diagnostics-13-00133-f006:**
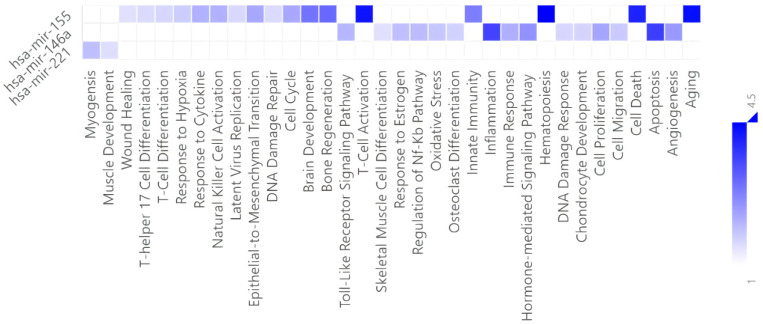
Significantly over-represented functional terms for miRNAs up-reulated amongst predicted targets in severe cases. The heat map shows that T-cell activation, inflammation, hematopoiesis, cell death and apoptosis and aging were the most enriched cell activities. False discovery rate (FDR) (*p* < 0.05).

**Table 1 diagnostics-13-00133-t001:** Selected sociodemographic characteristics, main comorbidities, and clinical symptoms for patients with severe COVID-19 (n = 41) and healthy controls (n = 42).

Variable	Category	Group, % (n)		*p*-Value *
		Controls	COVID-19	
Sex	Male	45.2 (19)	82.9 (34)	<0.01
	Female	54.7 (23)	14.6 (6)	<0.01
Age (years)	20–45	73.8 (31)	31.7 (13)	<0.01
	46–65	23.8 (10)	53.6 (22)	<0.01
	66–85	2.38 (1)	14.6 (6)	0.04
Current smoking		11.9 (5)	7.3 (3)	0.71
Type 2 diabetes (DM-II)		2.3 (1)	26.2 (11)	<0.01
Hypertension		9.5 (4)	41.4 (17)	<0.01
COPD or asthma		4.7 (2)	4.8 (2)	0.98
Immunosuppressed		0 (0)	2.4 (1)	0.49
Chronic kidney disease		0 (0)	4.8 (2)	0.24
Obesity, BMI ≥ 30 kg/m^2^		71.4 (30)	63.4 (26)	0.48
General symptomatology	Fever	2.3 (1)	73.1 (30)	<0.01
	Headache	16.6 (7)	58.5 (24)	<0.01
	Myalgia	16.6 (7)	63.4 (26)	<0.01
	Arthralgia	7.1 (3)	58.5 (24)	<0.01
Respiratory symptomatology	Cough	19.0 (8)	80.4 (33)	<0.01
	Odynophagia	9.5 (4)	31.7 (13)	0.01
	Dyspnea	7.1 (3)	85.3 (35)	<0.01
	Chest pain	11.9 (5)	36.5 (15)	<0.01
Other signs/symptoms	Anosmia/dysgeusia	2.3 (1)	4.8 (2)	0.61
	Diarrhea	7.1 (3)	24.3 (10)	0.03

COPD: chronic obstructive pulmonary disease; BMI: body mass index. Chi^2^ and Fisher’s exact tests were used. * *p* < 0.05 was considered statistically significant.

**Table 2 diagnostics-13-00133-t002:** Computed crude and adjusted odds ratios (OR) with 95% confidence intervals (CI) from logistic regression for the probability to identify severe COVID-19.

Variables	OR (95% CI)	
	Crude	Adjusted *
miR-16	1.28 (1.06–1.54)	–
miR-34a	1.12 (0.99–1.28)	–
miR-146a	0.70 (0.51–0.97)	0.24 (0.09–0.62)
miR-155	1.34 (1.13–1.60)	1.68 (1.19–2.37)
miR-221	1.29 (1.11–1.50)	1.36 (1.04–1.78)
Sex (male)	6.02 (2.16–16.7)	7.75 (1.40–42.7)
Age (years)	1.09 (1.04–1.14)	1.10 (1.04–1.18)
DM-II	15.1 (1.85–124.1)	-
HTN	6.83 (2.04–22.8)	-

DM-II: diabetes mellitus type II; HTN: hypertension; *Only significant variables (*p* < 0.05) remained in the final adjusted model; Nagelkerke R^2^ = 0.71, Hosmer–Lemeshow Chi^2^*, * p*-value = 0.53.

**Table 3 diagnostics-13-00133-t003:** Target genes validated in plasma/serum for the significant miRNAs.

Target Official Symbol	Official Full Name	Sequence Accession ID (Gene)	miRNA Associated with Regulation	Predicted/Validated	Gene Function	Reference
INPP5D (SHIP1)	Inositol polyphosphate-5-phosphatase D	NC_000002	hsa-miR-155-5p	Validated (qPCR, assay, luciferase reporter assay, and Western blot)	Tumor suppressor recognized to inhibit cell proliferation in many types of tumor cells.	[23,24]
CDKN1B	Cyclin-dependent kinase inhibitor 1B	NC_000012	hsa-miR-155-5p	Validated (qPCR, assay, luciferase reporter assay, and Western blot)	Plays a critical role in controlling cell growth and division. Macrophage proliferation.	[23]
SOCS1	Suppressor of cytokine signaling 1	NC_000016	hsa-miR-155-5p	Validated (qPCR, assay, luciferase reporter assay, and Western blot)	Acts as a negative feedback regulator to inhibit JAK2/STAT3 signaling. Control of systemic inflammation and promotes the proliferation and inflammation of macrophages through downregulating SHIP1.	[25]
FOXO3	Forkhead box O3	NC_000006	hsa-miR-155-5p	Validated (qPCR, dual-luciferase reporter system, and Western blot)	Regulatory effects on cell proliferation, apoptosis, metabolism, and oxidative stress. Plays an important role in both inflammation and regulation of cell proliferation. Regulates inflammation by NF-κB, T cells, and autoinflammation.	[26]
ICAM-1	Intercellular adhesion molecule 1	NC_000019	hsa-miR-155-5p	Validated (reporter gene assay, qPCR, and Western Blot)	Indicator of vascular inflammation.	[27]
PTEN	Phosphatase and tensin homolog	NC_000010	hsa-miR-221-3p	Validated (RT-qPCR)	Tumor suppressor by negatively regulating the AKT/PKB signaling pathway.	[28]
ICAM1	Intercellular adhesion molecule 1	NC_000019	hsa-miR-221-3p	Validated (RT-qPCR)	It binds to integrins of type CD11a / CD18, or CD11b / CD18, and is also exploited by rhinovirus as a receptor.	[29]
FOS	Fos proto-oncogene, AP-1 transcription factor subunit	NC_000014	hsa-miR-221-3p	Validated (RT-qPCR)	Implicated as regulators of cell proliferation, differentiation, and transformation.	[30]
NFKB1	Nuclear factor kappa B subunit 1	NC_000004	hsa-miR-221-3p	Validated (qRT-PCR)	Transcription regulator that is activated by various intra- and extracellular stimuli such as cytokines.	[29]
P27KIP1	Cyclin-dependent kinase inhibitor 1B	NC_000012	hsa-miR-221-3p	Validated (luciferase reporter assays, Western blotting, and qPCR)	p27kip1 is an inhibitor of cell-cycle progression. Plays a decisive role in nonproliferating cell types, such as eosinophils and dendritic cells (DCs).	[24]
EGFR	Epidermal growth factor receptor	NC_000007	hsa-miR-146a-3p	Validated (qPCR)	It is a receptor for members of the epidermal growth factor family. EGFR is a component of the cytokine storm, which contributes to a severe form of COVID-19 resulting from infection with severe acute respiratory syndrome coronavirus-2 (SARS-CoV-2).	[31]
SUMO1	Small ubiquitin-like modifier 1	NC_000002	hsa-miR-146a-3p	Validated (dual-luciferase assay, Western blot)	SUMOylation of SERCA2a, Ca^2+^ handling in cardiomyocytes.	[32]

## Data Availability

Data is contained within the article or Appendix A.

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
