# Peer review of "miR-146a, miR-221, and miR-155 are Involved in Inflammatory Immune Response in Severe COVID-19 Patients"

_diagnostics, 2022, doi:10.3390/diagnostics13010133_

Round 1
Reviewer 1 Report
1. How did the author predict and select the MicroRNAs?
2. What is the home message of the study?
3. Mention the limitation and future studies. Such as PMID: 35476290, PMID: 35476290
4. The author can analyze of roc cure for the viral microRNAs.
5. Draw a flowchart based on the PRISMA guideline.
6. The author believed that “Twelve of the predicted gene targets have been validated in plasma/serum, reflecting their potential use as predictive prognosis biomarkers.” In the laboratory, we can diagnose and prognosis COVID-19 via the validation genes and primers. Hence, what is the novelty of this study?
7. Roc cure is the vital analysis I think the author has to focus on in this section.
8. miR-155, mir-146a was evaluated before. Why did the author select these miRs?
Author Response
Dear reviewer
Thanks for all your comments. Surely, all these comments have contributed to improve the quality of the manuscript. Here you can find attached the point by point response.

Reviewer 2 Report
The manuscript "miR-146a, miR-221 and miR-155 are involved in inflammatory 2 immune response for severe covid-19 patients” is an interesting well-written research article that focuses on the identification of biomarkers involved in the pathological progression of COVID. However, I have some comments/ suggestions that may help improve the paper:
-To maintain the denomination of miR-146a during the text.
-To explain the reason to include miR-16 in this study. Applied biosystems recommend the use of this miR as endogenous control, but in some protocols, this miR is used as control of hemolysis.
-To note the limits in using the U6 as a control in plasma. U6 snRNA previously have been used to normalize circulating miRNAs in RT-qPCR analyses, but recently was demonstrated that this non-coding RNA has a lower stability and is less abundant than miRNAs in circulating. Moreover, this non-coding RNA does not reflect the biological properties of miRNAs and is highly expressed in tissues, but its expression in plasma is very low.
-In conclusion as preferable to indicate a correlation between total cholesterol and miR-221 in the same paraph there, you speak about this miR.
-I am not agreeing that the assessed ROC curve had excellent performance because, despite the fact as AUC is 0,95, the specificity of this curve is not high (about 0,6).
Author Response

(The authors gave the same response as above.)

Round 2
Reviewer 1 Report
The author did all my comments.